# Medical Imaging of Inflammations and Infections of Breast Implants

**DOI:** 10.3390/diagnostics13101807

**Published:** 2023-05-20

**Authors:** Elisabetta Giovannini, Laura Travascio, Giulia Anna Follacchio, Matteo Bauckneht, Benedetta Criscuoli, Camilla De Cataldo, Andrea Iozzelli, Andrea Cimini, Maria Ricci

**Affiliations:** 1Nuclear Medicine Unit, S. Andrea Hospital, 19127 La Spezia, Italy; 2Nuclear Medicine Unit, P.O. Spirito Santo, 65124 Pescara, Italy; lauratravascio.lt@gmail.com; 3Nuclear Medicine Unit, Macerata Hospital, AST, 62100 Macerata, Italy; 4Nuclear Medicine, IRCCS Ospedale Policlinico San Martino, 16132 Genova, Italy; 5Department of Health Sciences (DISSAL), University of Genoa, 16132 Genova, Italy; 6Department of Breast Imaging and Emergency Radiology, San Salvatore Hospital, 67100 L’Aquila, Italy; decataldocamilla@gmail.com; 7Radiology Unit, Macerata Hospital, AST, 62100 Macerata, Italy; 8Nuclear Medicine Unit, St. Salvatore Hospital, 67100 L’Aquila, Italy; 9Nuclear Medicine Unit, Cardarelli Hospital, 86100 Campobasso, Italy

**Keywords:** breast implant infections, mammography, ultrasound imaging, magnetic resonance imaging, scintigraphy, positron emission tomography

## Abstract

Breast implants are widely used for reconstructive and/or cosmetic purposes. Inflammations and infections of breast implants represent important complications in clinical practice. The proper management of complications is necessary: diagnostic imaging plays a key role in detecting sites of inflammation and/or infection. The present review aims to illustrate the radiological findings of these conditions with different imaging techniques, such as mammography (MX), ultrasound (US), magnetic resonance imaging (MRI), and nuclear medicine imaging. A knowledge of these findings is essential for radiologists and nuclear medicine physicians to provide helpful information for the clinical management of these complications.

## 1. Introduction

In breast cancer surgery the choice of reconstruction type is essential for patients and for pathology management. There are two major reconstructive pathways: autologous breast reconstruction and implant-based breast reconstruction [1]. Breast enlargement, asymmetries or breast anomalies correction and male-to-female top surgery also provide for the use of breast implants. Breast augmentation procedures with silicone breast implants are now a consolidated practice worldwide [2], with a total of 101,000 breast reconstructions and 314,000 augmentations performed in the United States alone as of 2018 [3].

However, despite the large number of women receiving breast implants, there are still many unresolved issues without a satisfactory amount of scientific evidence, including comparisons between different surgical options, the timing of breast implants relative to radiation and chemotherapy, implant materials, anatomic planes, and the use of human acellular dermal matrices (ADM) [4]. These topics are under active debate, given the increasing number of reconstructive breast surgery complications.

Adverse events after breast implants usually consist of local complications. The most common are pain, swelling, redness, infections, capsular contracture, implant rupture, and gel bleeding [5]. Implant complications should be identified early to avoid unnecessary and costly implant changes but, above all, to avoid consequential damage and more difficult retreatments. Infection is one of the most common complications, being reported in 1.9–2.5% of breast augmentations [6] and in 4.8–35.4% of breast reconstructions [7,8]. Acute infections show the typical signs and symptoms of cellulitis, i.e., breast pain, swelling, erythema and sometimes fever, eventually pus, and can progress to toxic shock syndrome within hours after surgery [9]; therefore, the diagnosis is generally clinical. In contrast, subacute or late infections can be subclinical. The majority of surgical-site-infection complications in immediate implant-based breast reconstructions occur more than 30 days after both the first-stage and second-stage procedures [10]. Subacute infections, several months after breast implantation, may appear with general malaise alone or with other symptoms (breast pain, movement of the prosthesis, or prolonged wound healing). Late infections occur months to years after implantation, with only vague breast pain, and may be revealed by infections at distant sites after the hematogenous seeding of the implant [9,10,11].

In recent years, acellular dermal matrices (ADMs) have been increasingly used as soft-tissue replacement devices in breast reconstructive surgery. ADMs are biologic mesh-like structures or matrices derived from dermal cells cleaned of cellular elements, and covered inside the breast pocket. Over time, the patient’s cells repopulate the ADM for full tissue integration. However, in some patients, the ADM does not integrate with the patient’s tissues, thus causing local inflammation and increasing the risk of implant failure/loss or the need for explant surgery [12].

Breast implant ruptures, silicone-induced granulomas, silicone abrasions, silicone bleedings, and capsular contractures have also been described [13,14,15,16].

Finally, although rare, oncological complications related to breast implants have been described (i.e., breast implant-associated anaplastic large-cell lymphoma, BIA-ALCL, a rare form of T-cell lymphoma, recently linked to the use of certain types of breast implants) [17,18]. All the cases mentioned above may benefit from the use of imaging tools to support a timely differential diagnosis. A knowledge of the corresponding radiological findings is essential for radiologists and nuclear medicine physicians to provide helpful information for the clinical management of these complications. On this basis, the present narrative review aims to summarize the the radiological findings for reconstructive breast surgery complications. An overview of the studies included in the present review is provided in Table 1.

## 2. Radiological Imaging

### 2.1. Mammography (MX)

On MX, breast implants are mostly depicted as regular-shape opacities, whose density depends on the used material. For example, mammographically translucent breast implants, made of soya-bean oil, were introduced onto the market in the past. Until recently, the use of saline implants was ordinary in the USA, which have a similar opacity to silicone implants. Double-lumen implants (silicon/saline) were also used in the past for esthetic purposes, and nowadays they are very common in patients with post-surgical oncology reconstruction with expanders. These types of implants usually are not recognizable as double lumen on MX.

The imaging features are variable, depending in part on the underlying cause that induced the infection or inflammation. Infections can be more commonly an early complication after breast augmentation or reconstruction. Radiological findings include: regular round or oval opacities, representing liquid or fluid collection; calcified irregular opacities, in this case referable to as granulomas; lobulated dense opacities, representing siliconomas with a surrounding inflammatory reaction; calcifications with or without mass opacity, related to parasitic infections; and asymmetrical dense fat tissue, with or without cutaneous thickening, related to associated mastitis. Asymmetrical dense fat tissue with cutaneous thickening is very commonly found in post-radiotherapy mastitis or post-surgical tissue reaction, both after a quadrantectomy or a mastectomy, and can persist in women who underwent breast implant reconstruction. Late infection is usually characterized by periprosthetic fluid collection, glandular edema, and cutaneous thickening [43].

Capsular contracture may mimic infections on MX, with specific features including the deformation of the implant that can asymmetrically increase the anterior-posterior size and thickening of the capsule, or the presence of calcifications that may improve the differential diagnosis.

Of note, mammographers are commonly being challenged to perform MX on patients with breast implants. Many clinicians and some radiologists erroneously assume that the presence of breast implants represents a limitation to the mammogram’s diagnostic value. In 1988, Eklund et al. [44] proposed a mammographic technique that consisted of maneuvers to posteriorly relocate the implant, so as to correctly examine the breast tissue from medio-lateral oblique and cranio-caudal views. The most posterior aspect of the breast is seen only on standard compression views, without posteriorly relocating the implant. However, in a comparison study of pre- and post-augmentation mammograms in the same patient, even with Eklund’s maneuvers, an amount of breast tissue (up to 19%) still could not be examined [45]. Some authors suggested that using all the possible MX projections (standard and with Eklund’s maneuvers) had a substantial advantage in identifying more breast abnormalities [46]. Moreover, in patients with large or extra-large breast implants and limited surrounding breast tissue, this technique cannot be performed, considering an increased risk of complications from the implants, such as rupture, due to mammography compression. In the past, some authors [47,48] demonstrated the diagnostic features of the xeromammography technique, for the representation of implants and adjacent breast tissue, in a negative mode and without pad compression, obtaining images with a very high level of detail. One of the advantages of this technique is that it relies on a better depiction of the tissue immediately surrounding the implant. However, it has been abandoned due to the high levels of ionizing radiation exposures.

### 2.2. Ultrasound

An ultrasound has many different advantages over other diagnostic techniques, being very fast, easily available, non-invasive, and relatively cheap. Moreover, in recent years, the introduction of other ancillary US techniques has significantly improved the diagnostic accuracy of the solely plain US examination: color-Doppler and power-Doppler, contrast-enhanced US, elastosonography, and, lately, artificial intelligence. On the other hand, breast US is critically operator-dependent, requiring a high learning curve and lengthy experience.

In US images, both saline and silicone implants appear to be internally anechoic, surrounded by a linear echogenic envelope, which can consist of a single or parallel echogenic lines. The internal material echogenicity can vary in the case of older implants or implant ruptures [49]. The implant shape usually consists of a regular oval or round shape with radial folds, called “ripples”, which are often palpable but do not represent an abnormal finding.

Basic US images can easily depict the presence of fluid, which can be complex with multiple internal echoes or simple and anechoic, such as in cases of infections, post-surgical seromas or hematomas, or any type of periprosthetic inflammatory reactions. Fluid collections surrounding implants are common non-specific findings, often simply due to a foreign-body reaction and chronic inflammation [49]. In some less common cases, fluid collection can be chronic and with complex echogenicity, especially after post-mastectomy reconstruction, requiring a cytology sample for diagnostic purposes [50].

Lobulated hypoechoic masses, such as siliconomas, implant capsule thickening, morphologic variations in implant shape, fat edema, or fat necrosis are easily identifiable. In cases of implant extracapsular ruptures, an US can show a characteristic “snowstorm sign” due to the silicone bleeding throughout the capsule [51].

Capsular contracture may not always have US findings, but sometimes just a thickening of the dense tissue around the implant can be identified [51].

Color-Doppler and power-Doppler [52], microvascular imaging [53], and contrast-enhanced ultrasound [54] allow a direct estimation of the presence of angiogenesis and hypervascularization, hence the presence of active inflammation in benign ultrasonographic alterations too. Elastosonography can estimate the presence and degree of fibrosis that may be present in the thickened periprosthetic tissue [55] and in any type of granulomatous hypoechoic masses.

### 2.3. MRI

Breast MRI is the most accurate technique for the evaluation of integrity in the case of the clinical suspicion of rupture and/or a suspicious or doubtful ultrasound picture for a prosthetic rupture, with a sensitivity and specificity varying between the different studies reported in the literature, which however is between 80 and 90% and 90 and 97%, respectively [56].

MRI diagnostic accuracy isreduced for asymptomatic patients, regardless of age or type of implant, making the use of MRI unjustified in the absence of symptoms.

Considering, however, its fundamental role in follow-ups for cancer patients, even in the case of mastectomy, it represents a versatile modality in case of mammoplasty, and is able to provide a great deal of information regarding the periprosthetic environment as well.

The study of prostheses requires a magnetic field with an intensity of at least 1.5 Tesla, a dedicated surface coil, high spatial resolution for the identification of the fine signs of rupture, and the use of sequences that differentiate among tissue components (silicone, water, and fat). The simultaneous presence of a suspected proliferative breast lesion may justify the administration of contrast medium, potentially improving the identification of inflammatory complications [56].

The possible complications of reconstructive or aesthetic mammoplasty may differ according to the type of breast augmentation materials and the time of onset.

Breast implants differ in the number of lumens (single or double-lumen) and in the filling material (silicone or saline).

Silicone implants show an intermediate-to-high signal on T2W images, a high signal on the silicone-specific sequence, and a loss of signal in the silicone-suppressed sequence, while saline implants demonstrate a high signal on T2W images. Both appear hypointense on T1W images.

Free filler injections represent another breast augmentation strategy, currently less used and in some cases even banned due to the high risk of migration. The most frequently encountered complication is the foreign-body reaction, with the formation of a granuloma with possible stromal fibrosis mimicking breast cancer.

Polyacrylamide gel (PAAG) injection has been used as a filling material in China and the Soviet Union since 1997. It was generally injected into the retro-glandular space where it organized itself into ‘cysts’.

Being more than 90 percent water, it appears as hyperintense on T2 sequences and hypointense on T1 sequences. In the case of over-infection, the presence of pus within the cysts results in an inhomogeneous signal on the T2 sequences and a restriction of diffusivity on the DWI sequences. Irregular and nodular peripheral enhancement is not uncommon in the post-contrastographic acquisition phases [57,58]. Free-silicone-gel injection is now illegal but still encountered in clinical practice, and leads to the formation of cysts with typical silicone-signal characteristics. Its complications include infections, inflammatory reactions, migration, embolism, and lymphadenopathy [59]. Autologous fat injection represents an alternative for additive mastoplasty that avoids the risk of foreign- body reactions; however, inflammatory reactions and adiponecrosis are very common.

Liquid paraffin injection, widely used for breast augmentation in the early 20th century, is now banned due to its serious adverse effects and potentially bad cosmesis.

Under normal conditions, the prosthetic shell is surrounded by a thin fibrous capsule that appears hypointense in all sequences; often concomitant is a small periprosthetic fluid collection that must be considered paraphysiological or as evidence of radial folds that must be considered as a kind of prosthetic adaptation to the surrounding tissues. In dynamic sequences with contrast medium administration, diffuse, fuzzy, or late periprosthetic impregnation can be observed as an expression of granulomatous inflammatory tissue. This finding should be noted, as it may sometimes require pharmacological treatment.

Early adverse events include hematoma and seroma, infection, changes in skin sensitivity, breast pain, and dysmetria, while late-onset capsular contracture and implant rupture may occur more commonly (Figure 1) [60,61]. Hot seroma is a fluid collection, frequently observed in the surgical bed, consensually to the scar, in the first weeks after surgery, showing heterogeneous signal intensity, and is generally intermediate-to-high on T2 sequences (Figure 2). If unresolved and infected it can result in an abscess, which is more frequently characterized by rim-enhancement, sometimes with an irregular and nodular profile.

Hematoma represents a less frequent periprosthetic finding, for which an MRI can provide information on the time of onset, with a hyperintense signal in the T1 sequences in the acute forms tending to decrease progressively in the subacute and chronic forms.

Among the different imaging techniques, MRI has the highest accuracy for the detection of peri-implant fluid collections or masses compared to mammography and ultrasound [61].

Breast implant-associated anaplastic large-cell lymphoma (ALCL) represents a further long-term complication. It is a rare form of T-cell lymphoma associated with breast- textured implants especially, arising generally 10 years after breast surgery. On an MRI it presents generally with a peri-implant effusion often associated with an enhancing mass and sometimes with ancillary lymphadenopathy. In these cases, a differential diagnosis with cold seroma, generally due to non-infective inflammatory processes, remains a diagnostic challenge. Late-onset periprosthetic fluid collection or evidence of a peri-implant mass requires cytological or histological verification in order to exclude this late complication.

## 3. Nuclear Medicine Imaging

For the study of breast implant complications, nuclear medicine procedures generally represent second-line tools after radiological imaging.

### 3.1. Scintigraphic Imaging

Scintigraphy can, in principle, be used to diagnose breast implant complications such as infection or inflammation secondary to capsular contracture, implant rupture, and granulomas. Since the widespread use of PET/CT imaging, scintigraphic imaging has been restricted to selected patients, as reflected by the few data currently available in the literature. Applications of scintigraphic procedures for the study of breast implant complications are limited to case reports illustrating either complex clinical conditions or incidental findings during scans requested for other causes. Among the various radiopharmaceuticals used in scintigraphic imaging, [67Ga] Ga-citrate, radiolabeled leukocytes, [99mTc] Tc-diphosphonates, and [131I] Iodine have been mentioned in these reports.

In past years, the radioisotope [67Ga] Ga-citrate was extensively used to study inflammatory and infectious processes due to its ability to bind to lactoferrin, activated leukocytes, and bacteria. A case report by Hartshorne et al. described the application of [67Ga] Ga-citrate scintigraphy to evaluate the inflammatory activity in a patient with a diagnosis of capsular contracture of a breast implant nonresponsive to medical treatment [42]. [67Ga] Ga-citrate scintigraphy revealed intense radiopharmaceutical uptake around the right breast implant, guiding clinicians to the surgical removal of the implant. [67Ga] Ga-citrate scintigraphy was repeated after surgery and verified the absence of significant uptake in the treated breast. The authors suggested that [67Ga] Ga-citrate scintigraphy could represent a useful tool to evaluate the presence and degree of inflammation in capsular contracture. A complex case was illustrated in 2000 by Leslie et al., concerning a transexual patient with renal failure, clinical signs of infection, a previous history of breast augmentation, and thoracic shingles [40]. [99mTc] Tc-diphosphonates scintigraphy was performed to evaluate a possible underlying rib osteomyelitis; no areas of bone uptake were detected but a faint soft-tissue tracer uptake around the left breast implant was noted. An evaluation proceeded with [67Ga] Ga-citrate scintigraphy which confirmed the presence of an enhanced tracer uptake in the same region, consistent with a peri-implant infection. The combination of these scintigraphic procedures contributed to solve a complex clinical condition and to offer the patient an accurate treatment. However, in recent years, [67Ga] Ga-citrate scintigraphy has been largely replaced by PET/CT and, at present, is proposed almost exclusively when PET/CT services are not available [19].

[99mTc] Tc-diphosphonates scintigraphy has been also used in a case of an incidental finding of breast implant rupture [40,41]. A bone scan was performed as a follow-up for metastatic breast cancer. A circular region of radiotracer uptake in the location of the patient’s left breast implant was seen in the planar images. Once a SPECT/CT was completed, the authors found that the patient’s breast implant had ruptured when compared to the prior CT. The [99mTc] Tc-MDP uptake in the capsule of the breast implant was attributed to rupture, likely secondary to inflammation.

Since its introduction in 1975, radiolabeled leukocytes’ scintigraphy has been increasingly used to diagnose infections due to its high specificity and sensitivity and its impact on medical and surgical management. Leukocyte radiolabeling techniques include, on the one hand, an in vitro multi-step procedure developed first using [111In] In-oxine and more recently with [99mTc] Tc D,L-HMPAO. On the other hand, radiolabeling can be obtained in vivo by using anti-granulocyte antibodies conjugated with [99mTc]Tc-diphosphonates. The first application of this procedure to diagnose a breast implant complication was reported by Ellenberger et al. in 1986 [41]. The authors described the use of [111In] In-oxine-labeled leukocytes scintigraphy in a patient with persistent Pseudomonas aeruginosa colonization after a bilateral breast implant removal due to infection. In their experience, radiolabeled leukocytes scintigraphy contributed to localized areas of persistent infection harbored around the retained polyurethane foam which previously covered the implant. The surgical excision of these retained infected areas was guided by radiolabeled leukocytes scintigraphy, providing a specific and reliable tool to ensure the complete recovery of the patient. At present, radiolabeled leukocytes scintigraphy still represents a valid diagnostic option for breast implant infections. Its strength is its high specificity, and should be considered as a second-line imaging tool for all cases in which a suspected infection needs to be disclosed in the presence of inflammatory findings in the first-line imaging.

Of note, breast implants’ tracer uptake on scintigraphic images may also be unspecific, thus potentially generating false-positive findings. In a recent case report, a 40-year-old woman underwent post-therapy scintigraphic imaging after receiving [131I] Iodine therapy for papillary thyroid cancer [20]. Postablation whole-body [131I] Iodine scintigraphy revealed not only increased activity in the thyroid bed but also in the anterior part of the chest. SPECT/CT images localized the activity to the bilateral breast implants. However, no other signs of breast implant complications were found during the subsequent diagnostic algorithm.

### 3.2. PET

Positron emission tomography/computed tomography (PET/CT) is currently performed to evaluate a variety of pathological processes by using specific molecules labeled with positron-emitting isotopes, including [18F] Fluorodeoxyglucose ([18F] FDG). Even if [18F] FDG PET/CT is generally requested for oncologic patients, [18F] FDG accumulation is not specific for neoplasms and occurs in a variety of benign inflammatory and infection processes. [18F] FDG is internalized in activated white blood cells, recruited in infected tissue through GLUT membrane transporters, and then phosphorylated to 2-deoxyglucose-6-phosphate and trapped in cells [21,62]. Indeed, whilst these foci of tracer uptake were initially considered to be pitfalls in oncologic scans, in recent years the [18F] FDG PET/CT imaging of infections has progressively gained in importance, contributing to the diagnosis and restaging of infectious diseases during and after therapy.

Investigating breast implant inflammation and infection by means of [18F] FDG PET/CT is reported mainly when breast prostheses are implanted in patients affected by breast cancer or in patients with symptoms and imaging consistent with lung (oncologic) afflictions (Figure 3 and Figure 4). Several objective findings, such as hard lumps under the skin around the implant, eventually with inflammation, rejections, contracture or, even worse, leakage and rupture of the implant, with silicone migration and distant granulomas are described in the literature [22,26,29,31,35,36,37,38,63] (Figure 5). Many cases manifest as focal tissue reaction or mediastinal or axillary lymph nodes pathology. Several authors reported images of intense [18F] FDG uptake in the axillary lymph nodes of patients with a rupture of a breast implant and a subsequent lymph node biopsy demonstrating benign inflammatory responses and no recurrence of malignancy [30,33,34,39,64,65].

Breast implant-associated anaplastic large-cell lymphoma (BIA-ALCL) and breast cancer recurrence have also been significantly associated with textured implants [23,24,25,28,32,66,67,68]. BIA-ACL is an infrequent T-cell Non-Hodgkin lymphoma causing breast asymmetry and swelling secondary to an effusion developing between the breast implant and the host fibrous capsule [27,69]. In advanced cases, it may involve lymph nodes related to the breast [70,71]. Recently, some rare cases of squamous cell carcinoma (SCC) [66] and lymphomas different from the BIA-ALCL have also been reported to FDA, which will collect them and give them a clinical identity and a role in patient’s outcome [72].

Nevertheless, inflammation, infection, and neoplastic foci may display similar levels of glucose uptake [73], and discriminating the etiology of each focus of uptake can be challenging, even bearing in mind the clinical history of each patient. In this circumstance, the use of semiquantitative measurements of tracer uptake, such as the standardized uptake value (SUV), cannot discriminate between neoplasm and its metastases from inflammation and infection. A dual-time-point PET, allowing the calculation of SUV changes from early-to-delayed [18F] FDG PET/CT scans, emerged as a promising method to overcome the poor [18F] FDG specificity for this differential diagnosis, thanks to the longer retention over time of [18F] FDG in neoplastic cells compared to other processes [74]. A CT embedded in PET/CT systems can also increase PET specificity by revealing specific morphological findings related to the infection in certain cases, i.e., gas bubbles in abscess formation [75]. Nevertheless, granulomatous benign lesions may show as well as an increasing [18F] FDG accumulation in late images and low-dose CT images without iodinated contrast are not informative in many cases. Accurate anamnesis, deep knowledge of pathophysiologic processes and, when needed, cultural or pathology confirmation can avoid patient mismanagement. In particular, every seroma or fluid collection around a breast prosthesis should be investigated and eventually cultured, as it is suspicious of an infection.

Other non-FDG radiopharmaceuticals may play a role in breast implant infections. [18F] Choline has been described as being taken up in inflammatory processes [74]. Moreover, we may speculate that other proliferation tracers, including [11C] Methionine or [18F] fluorothymidina, may play a role in breast implants complication’s imaging [75]. Finally, [68Ga]-labeled fibroblast-activation-protein inhibitors (FAPI) may track chronic infections, as fibroblast-activated protein is expressed in the cells of the microenvironment [76].

## 4. Discussion and Conclusions

Since the number of breast implant procedures is increasing and an increasing number of patients are present for assessing implant integrity, imaging specialists should be familiar with the spectrum of appearances of these complications (inflammations, infections, and rupture of breast implants) (Table 2). The timely recognition of these situations is important to avoid irreversible damage, requiring further workup and surgery.

The imaging appearances of common breast implants and their complications are varied. MX has a limited role in their its evaluation, as inflammations, infections and capsular contracture may present superimposable MX findings. Indeed, its use can be proposed as a first-level option for suspected breast implant complications. The technical limitations of MX hamper its diagnostic power and dedicated procedures, potentially able to improve MX applications in this field (i.e., Eklund’s maneuvers), cannot be performed in all patients. For all cases in which MX cannot be performed or it results are inconclusive, the greatest diagnostic value lies in US and MRI that can be proposed as second-line imaging tools.

US allows a more detailed evaluation of the implant, allowing the identification of intracapsular ruptures and nonspecific inflammation signs, including fluid collections surrounding implants and hypervascularization. Furthermore, US imaging is ideal for invasive breast diagnostic procedures such as core biopsy and fine-needle aspiration biopsy, to obtain cytologic or histologic samples of undetermined or suspicious US findings.

MRI currently represents the technique with a greater sensitivity and specificity for the evaluation of silicone breast implant integrity and the identification of inflammatory complications. Its diagnostic value is higher in symptomatic than asymptomatic patients and when contrast media is administered.

As demonstrated by the few studies available in the literature, nuclear medicine methods play a limited role in the diagnostic algorithm for suspected breast implant complications. These studies mainly include case reports illustrating either complex clinical conditions or incidental findings during scans requested for other causes. Two exceptions are represented by PET/CT imaging and radiolabeled leukocytes scintigraphy. These technologies may represent valid second-line imaging tools in case of suspected breast implant infections, thanks to their additive role in sensitivity (the former) and specificity (the latter).

Many of the studies we reviewed were conducted with a small sample of patients. While these samples are still useful for providing functional and beneficial information, we believe that given the amount of medical data collected globally and today’s technology, it should be more feasible to produce studies with a much larger patient base.

Therefore we encourage the collaboration of multiple centers and the use of clinical data interoperability, artificial intelligence, and big data processing to aid the data collection process for similar future studies.

## Figures and Tables

**Figure 1 diagnostics-13-01807-f001:**
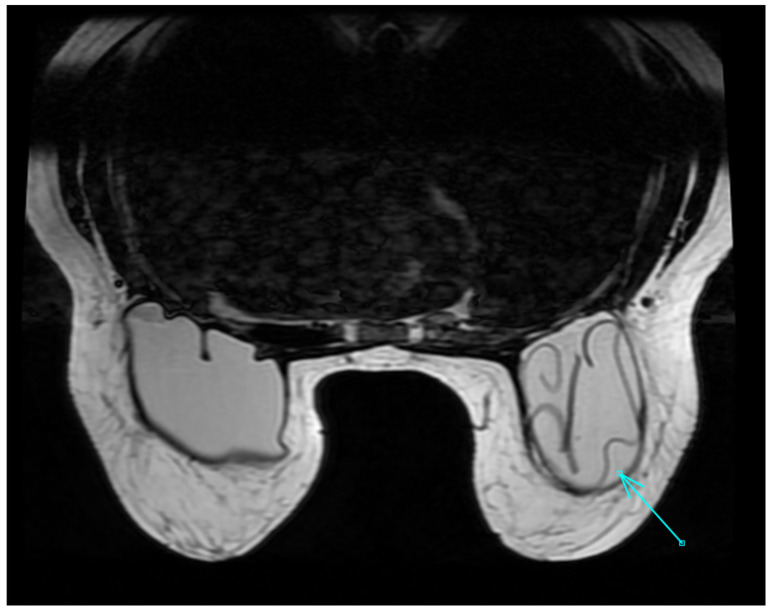
MRI image of a 47-year-old woman with bilateral reconstructive implants after surgery for breast cancer. *Linguine sign:* proof of intracapsular rupture. The curvilinear lines (arrow) are formed by the ruptured envelope and look like *Linguine* pasta.

**Figure 2 diagnostics-13-01807-f002:**
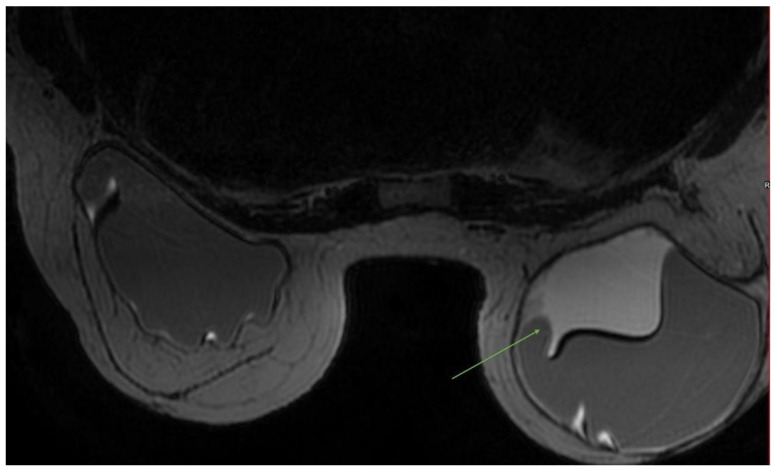
MRI image of a 49-year-old woman with a history of bilateral reconstructive implants after breast surgery for breast cancer. Axial T2-images show a voluminous seroma, adjacent to the right prosthesis (green arrow).

**Figure 3 diagnostics-13-01807-f003:**
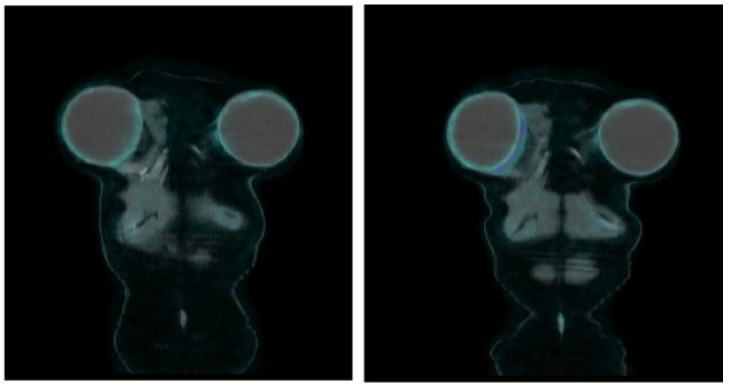
Inflammatory reaction from a partial prosthetic mobilization. A 47-year-old woman in for a follow-up for breast cancer with a history of reconstructive bilateral implants after a bilateral mastectomy and a recent diagnosis of partial right prosthetic mobilization. The FDG PET/CT image on the left, a re-staging PET study after neoadjuvant therapy and surgery performed a few months before the diagnosis of prosthetic mobilization, shows the symmetrical distribution of the tracer in the periprosthetic tissues bilaterally. The FDG PET/CT image on the right, performed 5 months after the previous scan, reveals the appearance of diffuse and moderate FDG uptake in the medial and inferior sides of the right breast prosthesis, a site of partial prosthetic mobilization, suggesting an inflammatory reaction of the periprosthetic tissues.

**Figure 4 diagnostics-13-01807-f004:**
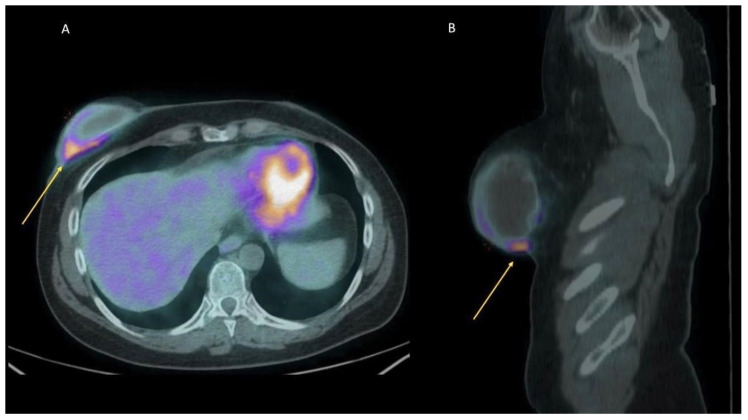
Breast implant infection. A FDG-PET/CT for a 51-year-old woman with a breast implant infection. Axial (**A**) and sagittal (**B**) images show an area of severe increased uptake of the radiopharmaceutical (yellow arrows, SUV max 6.7) adjacent to the lower region of the right breast implant, site of the infection process.

**Figure 5 diagnostics-13-01807-f005:**
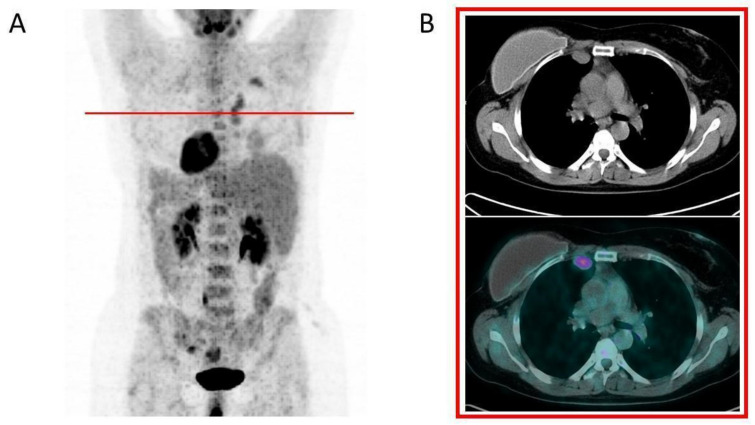
Iatrogenic intrathoracic encapsulated siliconomas from a ruptured breast implant. A 45-year-old woman with a history of right silicone-based reconstructive implant after breast surgery for breast cancer. Several years later the patient began to experience progressive fatigue. Restaging with FDG PET/CT revealed the presence of FDG-active supraclavicular and internal mammary lymphadenopathies, as clearly depicted in maximum intensity projection image (**A**) and in axial CT and PET/CT images (**B**). She underwent a biopsy that confirmed the presence of silicone granulomas. The red line corresponds to the transaxial plane.

**Table 1 diagnostics-13-01807-t001:** Overview of the main studies included in this review concerning nuclear medicine technique. Studies are listed by year from newest to oldest.

Authors	Publication Year	Nuclear Medicine Imaging Technique	No. of Patients	Main Findings
Hudson A et al. [19]	2022	[99mTc]Tc-diphosphonates Scintigraphy	1	Implant rupture
Wang Y et al. [20]	2022	131Iodine Scintigraphy	1	Aspecific implant uptake
Vedala K et al. [21]	2021	[18F]FDG PET/CT	1	Granulomatosis
Khakbaz E et al. [22]	2021	[18F]FDG PET/CT	1	Granulomatosis
Verde F et al. [23]	2020	[18F]FDG PET/MRI	1	BIA-ALCL
Pandika V et al. [24]	2020	[18F]FDG PET/CT	4	BIA-ALCL
Mescam L et al. [25]	2020	[18F]FDG PET/CT	3	BIA-ALCL
Phan S et al. [26]	2020	[18F]FDG PET/CT	1	Granulomatosis
Montes Fernandez M et al. [27]	2019	[18F]FDG PET/CT	1	BIA-ALCL
Siminiak N et al. [28]	2019	[18F]FDG PET/CT	2	BIA-ALCL
Palot Manzil FF et al. [29]	2018	[18F]FDG PET/CT	1	Granulomatosis
Dominguez ML et al. [30]	2018	[18F]FDG PET/CT	1	Granulomatosis
D’hulst L et al. [31]	2016	[18F]FDG PET/CT	1	Granulomatosis
Acevedo-Banez I et al. [32]	2015	[18F]FDG PET/CT	1	BIA-ALCL
Karnatovskaia LV et al. [33]	2014	[18F]FDG PET/CT	1	Nodular Lymphoid Hyperplasia
Ulaner GA et al. [34]	2013	[18F]FDG PET/CT	1	Granulomatosis
Soudack M. et al. [35]	2013	[18F]FDG PET/CT	12	Granulomatosis
Ho L et al. [36]	2010	[18F]FDG PET/CT	1	Granulomatosis
Chen C et al. [37]	2009	[18F]FDG PET/CT	1	Granulomatosis
Bhargava P et al. [38]	2006	[18F]FDG PET/CT	1	Capsular rupture
Hurwitz R et al. [39]	2003	[18F]FDG PET/CT	1	Granulomatosis
Leslie K et al. [40]	2000	[67Ga] Gallium-citrate Scintigraphy	1	Implant infection
Ellenberger P et al. [41]	1985	[99mTc] Tc-HMPAO-labeled leukocytes Scintigraphy	1	Implant infection
Hartshorne MF et al. [42]	1982	[67Ga] Gallium-citrate Scintigraphy	1	Capsular contracture

[18F]FDG: [18F]Fluorodeoxyglucose; PET/CT: Positron Emission Tomography/Computed Tomography; PET/MRI: Positron Emission Tomography/Magnetic Resonance Imaging; BIA-ALCL: Breast implant-associated anaplastic large-cell lymphoma.

**Table 2 diagnostics-13-01807-t002:** Approach to evaluate the complications of breast implants.

Imaging Technique	Normal Findings	Abnormal Findings	Comments
Mammography (MX)	- Regular-shape opacities; - Density dependent on the used material, up to radiolucent in liquid material.	- Regular round or oval opacities; - Double-implant contour, sign of pericapsular fluid collection; - Calcified irregular opacities, sign of granulomas; - Lobulated dense opacities, sign of siliconomas with surrounding inflammatory reaction; - Calcifications with or without mass opacity, signs of parasitic infections; - Asymmetrical dense fat tissue, sign of mastitis, with or without cutaneous thickening; - Cutaneous thickening, common sign of mastitis and/or after radiosurgery; - Periprosthetic fluid collection, glandular edema, and cutaneous thickening, signs of late infection; - Capsular contracture may mimic infection, specific signs of contracture include implants deformation, capsular thickening, and presence of calcifications.	Worldwide diagnostic technique for breast assessment, but in case with breast implants the accuracy is reduced. Combination of standard and projections with Eklund’s maneuvers increases the diagnostic accuracy. Not possible in patients with large or extra-large breast implants.
Ultrasound (US)	-Regular, linear echogenic implant wall, oval or round in shape; -A second chamber is always found in breast expanders and in double-lumen implants; -The peri-prosthetic capsule is depicted as two parallel echogenic lines; -Implant wall folding “ripples” can be normally present as regular wall waves; - Minimal layers of peri-capsular hypoechoic liquid can be normally present; -A single round regular interruption of the parallel lines consists in the valve, present in all breast expanders on the upper-external side. A single-lumen implant valve is positioned on the posterior side and usually not visible on US.	- Abnormally echoic or abundant peri-prosthetic fluid collection is a sign of inflammation, infection, or implant rejection; - Large peri-prosthetic focal seromas or hematomas can be commonly found in the immediate post-surgical period; -Signs of capsular contracture such as inhomogeneous thickened capsule and irregular ripples are uncommonly visible on US; - Siliconomas are shown as hyperechoic regular/oval-shaped nodules; - Fat edema or fat necrosis can be identifiable on US as signs of liponecrosis, infection, or post-radiation changes; - The “snowstorm sign” is a rare but typical sign of extracapsular rupture; - Intracapsular rupture can be seen as regular hyperechoic intra-prosthetic lines; - Capsular hypervascularization is always a sign of active inflammation or malignancy; - Contrast-enhanced ultrasound (CEUS) can improve the demonstration of hypervascularization; - Elastosonography can detect the presence and estimate the degree of capsular fibrosis, for example in capsular contracture; - In the differential diagnosis between infection and inflammation, US-guided fine-needle aspiration biopsy (FNAB) can play a crucial role.	Widely available diagnostic technique, considerably operator-dependent: a high level of expertise in the evaluation of breast implant abnormalities and breast focal lesions is required. Ancillary techniques, such as color/power-Doppler, US contrast agent administration, and elastosonography, can significantly improve the diagnostic accuracy. US is the most simple and affordable tool to be used as a guide for diagnostic invasive procedures, such as FNAB and core biopsy.
MRI	Breast implants show different intensity signals due to their composition: -Silicone single lumen has an intermediate-to-high signal on T2W images, a high signal on the silicone-specific sequence, and a loss of signal in the silicone suppressed sequence; -Saline, single lumen has a high signal on T2W images; -Standard double lumen (outer saline, inner silicone); -Reverse double lumen (outer silicone, inner saline). A fibrous capsule hypointense in all sequences and a small periprosthetic fluid amount are paraphysiological findings.	Acute complications: -Hematoma —hyperintense on T1W images, decreasing over time; -Seroma—intermediate-to-hyperintense on T2W images; -Abscess—fluid collection with irregular, thick peripheral enhancement; -Ancillary signs—edema, skin thickening, and adenopathy. Late complications: -Capsular contraction—prosthetic contour alterations, peripheral enhancement; -Intracapsular breast implant rupture (uncollapsed rupture “keyhole sign”, minimal collapse “subcapsular line sign”, and partial-to-full collapse “linguine sign”); -Extracapsular breast implant rupture; -Rare, breast implant-associated anaplastic large-cell lymphoma, ALCL (peri-implant collection with an enhancing mass and lymphadenopathy).	Breast magnetic resonance imaging is the most accurate technique to assess prosthetic integrity in the clinical or ultrasound suspicion of rupture, but is not justified as a pure screening examination in asymptomatic women of all ages and with any type of prosthesis. Its parametric nature allows the typing of the content of periprosthetic fluid collections (seroma, hematoma) and, combined with the administration of contrast medium, the detection of periprosthetic neoplastic recurrences or complications (breast implant-associated anaplastic large-cell lymphoma, ALCL).
[67Ga]Ga-citrate Scintigraphy	-No uptake around the implant.	-Different degree of radiopharmaceutical uptake in inflammatory/infected foci.	Since the introduction of [18F]FDG PET/CT, [67Ga]Ga-citrate scintigraphy can be proposed where PET/CT is not available.
Radiolabeled leukocytes’ Scintigraphy	-No uptake around the implant.	-Increasing uptake over time in areas of leukocyte-mediated infection.	Radiolabeled leukocytes scintigraphy still represents a possible diagnostic option for breast implant infections and should be considered as a second-line imaging tool in cases that remain equivocal after first-line imaging.
[18F]FDG PET/CT	-No uptake or only faint uptake around the breast implant. -No axillary lymph node uptake, or just faint uptake in normally-sized nodes, vascular hilum well-visible.	-Focal uptake around the implant and in axillary, mediastinal (usually internal mammary), and supraclavicular enlarged lymph nodes; -Pericapsular fluid collection may be present, with detectable faint activity; -Fluid effusion between the breast implant and the host fibrous capsule causing asymmetry and swelling of the breast can be a sign of breast implant-associated anaplastic large-cell lymphoma.	Even bearing in mind the clinical history of each patient, both visual and semiquantitative analysis (SUVmax) do not discriminate among inflammation, infection, and neoplastic foci, because they take up glucose similarly. The clinical setting of each focal uptake (implantation for oncological versus aesthetic reasons) and any morphological findings (see above) may lead the clinician to follow-up or to collect a biopsy specimen, and eventually fluid culturing, to rule out granuloma/infection versus node metastases or lymphoma or SCC associated with breast implants.

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
