# Peer review of "Medical Imaging of Inflammations and Infections of Breast Implants"

_diagnostics, 2023, doi:10.3390/diagnostics13101807_

Round 1

Reviewer 1 Report

Dear authors, I read with interest the article titled medical imaging of  infection and inflammation of breast implants.

However the article does not eloquently describe any salient features  or highlight a specific feature in any of the modality to utilize as a parameter to conclude of either an infection or inflammation. The manuscript touchs upon few points which can be considered as possibilities.

Instead if a table were created for each modality - radiology and nuclear medicine (NM) mentioning certain imaging characteristics favoring either of the pathologies it would be a good comparator. The various radioisotopes used in NM can be enumerated and findings that are characteristic can be demonstrated through images in available or description.  

Author Response

Thank you very much for your precious suggestions.

We have created a table (Table 2 Approach to evaluate the complications of breast implants) where the various diagnostic methods and the main pathological finding descriptions have been included. For Mammography, MRI and ultrasound we reported more information than nuclear medicine imaging in agreement with the literature. 

Reviewer 2 Report

It's an interesting ans well-written manuscript. I have a few suggestion.

It should be good to have more iconographies: for mammography, SPECT/CT and ultrason.

I also suggest having an iconography on the PET/CT showing the hypermetabolism around the implants.

Author Response

Thank you very much for your precious suggestions.

We have inserted some images regarding MRI and PET in patients with common prosthetic and periprosthetic complications.